# Cultivation potential of the tropical carrageenophyte *Eucheumatopsis isiformis* (Solieriaceae, Rhodophyta) from Yucatán, Mexico

**Monserrat López-Yllescas**[1], **Erika F. Vázquez-Delfín**[1], **Adrián Fagundo-Mollineda**[1¤], **Yolanda Freile-Pelegrín**[1], **Raquel Muñiz-Salazar**[2], **Loretta M. Roberson**[3], **Daniel Robledo**[1*]

**1** Centro de Investigación y de Estudios Avanzados del Instituto Politécnico Nacional, (Cinvestav), Unidad Mérida, Applied Phycology Laboratory, Mérida, Yucatán, Mexico, **2** Universidad Autónoma de Baja California (UABC), Laboratory of Epidemiology and Molecular Ecology, School of Health Sciences, Campus Ensenada, Ensenada, Baja California, México, **3** Marine Biological Laboratory (MBL), Woods Hole, Massachusetts United States of America

¤ Current address: Spanish Bank of Algae (BEA), Instituto de Oceanografía y Cambio Global (IOCAG), Universidad de Las Palmas de Gran Canaria, Gran Canaria, España

* daniel.robledo@cinvestav.mx

## Abstract

The production of tropical red seaweeds, collectively referred to as eucheumatoids, has stagnated or declined over the past decade. This trend is primarily attributed to limited availability of high-quality planting material, increased incidence of diseases, and insufficient genetic exchange between cultivated stocks and wild populations. Consequently, enhancing species diversification in commercial seaweed farming through the controlled management of reproductive strategies remains a significant global challenge. *Eucheumatopsis isiformis* is a native eucheumatoid from the tropical Atlantic Ocean that contains a high proportion of iota-carrageenan with a kappa-iota-nu carrageenan hybrid structure and represents a promising candidate for diversification of commercial crops. This study reports the first successful induction of sporulation in this species, thereby expanding current knowledge of its reproductive phenology and supporting its potential cultivation from spores. Molecular analyses confirmed the identity of *E. isiformis* across the three morphotypes examined, which exhibited comparable carrageenan yields. Collectively, these findings establish a robust methodological foundation for the development of spore-based production systems for this species. Sporulation was successfully induced after 5 and 10 days of acclimatization, with the highest carpospore densities obtained under treatment with putrescine at $10^{-3}$ M. Optimal spore development was observed at a light intensity of 40 µmol·m$^{-2}$·s$^{-1}$. These findings highlight the potential for advancing the cultivation of *E. isiformis* through consolidation of selective breeding and domestication protocols, which may also be applicable to other eucheumatoid species.

**Data availability statement:** All relevant data are within the paper and its Supporting information file.

**Funding:** This study was funded by the project ARPA-e DE-AR0000912 - Sub-award 54336 Development of Techniques for the Cultivation of Tropical Algae. The funders had no role in study design, data collection and analysis, decision to publish, or preparation of the manuscript.

**Competing interests:** No authors have competing interests.

## Introduction

Seaweed production has increased globally from 12 million tons in 2000–38 million tons in 2022 [1]. However, in the case of Eucheumatoids, a decrease in biomass production since 2013 has been associated with low genetic variability in seed stock, fluctuating environmental conditions, and extreme climatic events, and inadequate or inconsistent biosecurity legislation and policy frameworks [2]. To overcome the demand for carrageenan from eucheumatoids, genetic diversification from wild reproductive material has gained interest to improve its production [3–5]. In addition, other species with cultivation potential have been considered, including the native eucheumatoid *Eucheumatopsis isiformis* [6–14].

In seaweed breeding and nursery systems, genetic exchange plays a critical role in shaping both the genotype and phenotype [15–17]. Therefore, understanding reproductive phenology, phylogeography, and genomic adaptation of wild seaweed populations to long-term local environmental variability is essential for the development of sustainable mariculture practices [18–22]. Research on seasonal and interannual biochemical and physiological changes provides critical insights for optimizing cultivation strategies under changing climatic conditions. In addition, the use of molecular markers in the selection, improvement, and conservation of strains is essential for enhancing the resilience and productivity of the seaweed industry [23,24].

Genetic information has important systematic [25–28] and ecological implications that are closely linked to the life cycle of seaweed [20,29,30]. In eucheumatoids, the life cycle follows the typical triphasic life cycle characteristic of red algae, although some species exhibiting secondarily derived modifications [31–33]. Studies addressing the reproduction of *E. isiformis* are scarce [34,35], with most reproductive research focusing on commercially cultivated *Eucheuma*/*Kappaphycus* species [36]. Additional research has described the morphology and anatomy of *E. isiformis*, including its reproductive structures, mainly for taxonomic and ecological purposes [34,35,37–40]. In contrast, experimental induction of spore release using exogenous inducers, has primarily been explored in other red seaweed taxa [41–47].

This study aims to assess the reproductive capacity of *E. isiformis* by analyzing the morphological and anatomical characteristics of its morphotypes native to the Yucatan coasts in the southern Gulf of Mexico. In addition, the study provides essential information for spore based cultivation of *E. isiformis* using exogenous inducers, supporting efforts toward species diversification, selective breeding and the sustainable mariculture of eucheumatoids.

## Materials and methods

### Specimen collection

Specimens of *Eucheumatopsis isiformis* were collected at Dzilam de Bravo, Yucatán, México (21° 23.861'N, 88° 54.11'W) at depths of 2–4 m. Collection does not require a license or permit to harvest seaweed by hand for scientific use. Reproductive female gametophytes bearing cystocarps were collected bimonthly from March to November

2022 and in March 2023 for morphological and anatomical characterization at the Applied Phycology Laboratory, Centro de Investigación y de Estudios Avanzados del Instituto Politécnico Nacional, Unidad Mérida. Additional specimens representing different life cycle stages were collected monthly from March to June and August 2024, and February to March 2025, to assess reproductive stages and conduct sporulation experiments. Samples were transported to the laboratory in hermetic plastic bags under low temperature using coolers. For molecular analysis, specimens of *E. isiformis* collected between September and December 2023 (13 specimens) and in March 2024 (16 specimens), along with laboratory strain of *Kappaphycus alvarezii* (LabMEX), were dried in silica gel and subsequently preserved in vacuum-sealed bags. Molecular analyses were performed at the Laboratory of Epidemiology and Molecular Ecology, Universidad Autónoma de Baja California, Mexico.

## Molecular analysis

Genomic DNA was extracted from approximately 0.02 to 0.04 g of dry tissue using a CTAB/PVP (hexadecyltrimethylammonium bromide/polyvinylpyrrolidone) protocol [48] with modifications [49]. DNA concentration was quantified using a Nanodrop spectrophotometer, diluted to 50 ng/μL, and stored at –20 °C until further analysis. Three genetic markers for *cox*1 [50], *rbc*L [51], and RuBisCo spacer [52] were amplified under identical PCR conditions. PCR reactions (25 mL) contained 0.1–0.2 mg DNA template, PCR buffer (1 mM Tris-HCl; pH 8.3; 1.5 mM $MgCl_2$; 50 mM KCl; 0.02 mM each dNTP), 10.0 pmol of each forward and reverse primer, 0.3 units of Taq polymerase (Kappa), and 2 mg/mL of bovine serum albumin. PCR products were electrophoresed on 2.0% agarose gel stained with GelStar, run at 100 V for 40 min, and visualized under UV light. Amplified products were purified and sequenced in both forward and reverse directions using the same primers on Applied Biosystems 377 sequencers (Applied Biosystems, Foster City, California, USA). For quality-control, one individual per morphotype was re-extracted and reanalyzed. Sequence electropherograms were examined for quality and edited using Codon Code Aligner [53]. All sequences were aligned using CLUSTAL [54] and analyzed in MEGA5 [55]. Sequences of *Eucheumatopsis isiformis* and its synonym *Eucheuma isiforme* [44], retrieved from GenBank, were used as the outgroup for all three loci analyzed. All characters were equally weighted, and gaps were treated as missing data. Pairwise genetic divergence differences within and between species was assessed using MEGA X [56]. The best fitting evolutionary model was selected according to the Akaike information criterion [57] using jModelTest v2.1.17 [58]. A Bayesian phylogenetic tree was inferred from a concatenated dataset of *cox*1 + *rbc*L + RuBisCo spacer sequences using BEAST 2.5 [59], applying a relaxed molecular clock, the Yule speciation model, and the 012343 + I + G + F mutation model. The analysis ran for 10 million MCMC steps, yielding effective sample sizes (ESS) well above 200 for all parameters. Trees were visualized using FigTree v1.4 software [60].

## Carrageenan extraction

The carrageenan content of *E. isiformis* morphotypes collected in 2022 was analyzed. Both native and alkali-treated carrageenan were extracted (*n* = 3) using microwave-assisted extraction (MAE) with a Microwave Accelerated Reaction System (MARS6, CEM Company, USA) following the protocol described by [61]. Briefly, dry seaweed was rehydrated at room temperature for 12 h in 1% KOH solution (1:50 w/v). The sample was then heated in a closed vessel system (OMNI XP 1500). Microwave thermal program included a 5 min heat-up followed by a 10 min extraction time at 105 °C. After extraction, the solution was mixed with diatomaceous earth (Celite), filtered, and the filtrate neutralized to pH 8.5–8.6 using 10N HCl. Carrageenan was precipitated by the slow addition of 2% Cetavlon (hexadecyl-tri-methylammonium bromide) in a 9:1 (v/v) distilled water/acetone solution and recovered on filter paper. The fibrous carrageenan was washed three times with 95% ethanol saturated with sodium acetate to remove residual Cetavlon, followed by three additional washes with 95% ethanol to remove sodium acetate. Samples were then dried at 60 ºC for 24 h and weighed to calculate percent yield. For native carrageenan extraction, the same procedure was used, except that the initial KOH treatment was omitted.

## Morphological and anatomical characterization

Specimens of *E. isiformis* collected during 2022 and 2023 were cleaned and characterized. For each specimen, morphological and anatomical traits were recorded, including thallus height, branching pattern, branch shape, length and diameter, apex shape, presence and morphology of spines or branchlets, spine length, number of branching orders, presence of reproductive structures, and cystocarp density. Transverse and longitudinal sections of axes were prepared to measure branch diameter. For comparison, specimens identified as *E. isiformis* from Bahia Honda, Florida, USA, were also characterized.

In the mature female gametophyte, cystocarps were anatomically described and classified according to their maturation stage. Photographs were taken using a stereoscope microscope (Leica Zoom 2000) and optical microscopes (Leica DM 1000 LED and Zeiss Primostar). To determine cystocarp maturation, measurements included cystocarp diameter (mm), pericarp thickness (µm), carposporangia length and width (µm), and carpospore diameter (µm). For pericarp thickness, at least 30 cystocarps per specimen were measured at the mid-region, before the ostiole. Methylene blue and Alcian blue were used to stain preparations.

Cystocarp density was quantified by counting the number of cystocarps per thallus and per branch from March to November 2022. The number of cystocarps per 10 cm of branch was calculated. In March 2023, analyses of cystocarp density, distribution, and ostiole characteristics were conducted on three 1st-order branches per specimen. Ostioles were classified as protuberant, distinguishable open or opening, developed but pigmented, closed, or deformed/broken.

For specimens in the tetrasporophyte phase, branch length and diameter were measured. Transverse and longitudinal sections of axes were used to determine branch diameter and branching order. Tetrasporangia length and width were measured using a Zeiss Primostar optical microscope with a Zeiss Axiocam 208 color camera and images recorded with Zeiss ZEN 3.9 software.

## Sporulation experiments

Fertile carposporophytes of *Eucheumatopsis isiformis* collected in 2024 and 2025 were used for sporulation induction experiments using exogenous inducers. Specimens were transported to the laboratory at low temperature and cleaned using a soft bristle brush, 5% alcohol, and sterile seawater. Fragments were then acclimatized for 5–10 days in a Thermo Scientific Precision refrigerated incubator at 23°C and 25°C, 30 PSU salinity, 10−15 µmol·m$^{-2}$·s$^{-1}$ irradiance, 8:16 h light:dark photoperiod, and constant aeration. These conditions were determined based on previous studies [41–47] and preliminary acclimatization tests. Sterile seawater was renewed every two days, and specimens were maintained under these conditions throughout acclimatization.

For all the activities described below, all materials used for handling reproductive specimens and cystocarps were sterilized. Seawater was filtered through a 5 µm water filter (Pentair P5), subjected to ultraviolet sterilization (Tropical Marine Center P6-330W), and autoclaved at 15 PSI for 15 min along with all cultivation instruments. Materials that could not be autoclaved were exposed to ultraviolet light for 15 minutes. Brushes were sterilized with 5% alcohol, and tweezers were sterilized with alcohol and flame immediately prior to use.

Cystocarps of each specimen were examined using a stereoscopic microscope (Leica Zoom 2000). Transverse sections were made of five randomly selected cystocarps per specimen to verify the presence of characteristic maturation structures, including central fusion cells surrounded by diploid candelabra-like gonimoblasts with peripheral carposporangia and filaments extending into the pericarp. Cystocarps with distinguishable ostioles that were open or opening, without visible sporulation, or containing only a few carpospores were selected under an optical microscope (Zeiss Primostar). Fragments of 2.5 cm with mature cystocarps (1–2.5 mm diameter) were chosen for spore induction experiments.

Selected fragments were sterilized by first immersing them in a solution of 10 mL iodine per 100 mL sterile distilled water for five minutes, followed by immersion in a solution of 0.5 mL chlorine per100 mL sterile distilled water for five seconds. Finally, the fragments were placed in a solution containing 100 mg Benzyl Penicillin G and 250 mg of Ampicillin

per100 mL sterile seawater for 12 hours. After the antibiotic treatment, fragments with cystocarps were placed in 15 mL tubes containing 5 mL of Provasoli culture medium (PES 4%), with three replicated per treatment. Exogenous polyamine inducers, putrescine (PUT) and spermine (SPM), were applied at two concentrations ($10^{-3}$ and $10^{-6}$ M), while PES 4% served as control. Tubes were maintained in a Thermo Scientific Precision Incubator at 23°C, photoperiod 8:16 h (light:dark), a salinity of 30 PSU, and three irradiance levels, 0 (dark), 20, and 40 µmol·m$^{-2}$·s$^{-1}$. Conditions were optimized for *E. isiformis* based on previous studies in other red seaweed [62–65]. Each fragment was transferred to fresh culture medium every 24 hours for five consecutive days, and the number of released carpospores was recorded daily. To determine final spore density, 1 mL aliquots from each tube were homogenized and counted using a Sedgewick-Rafter counting chamber. Photographic documentation was performed with a Zeiss ZEN 3.9 software using a Zeiss Primostar optical microscope equipped with a Zeiss Axiocam 208 color camera.

## Statistical analysis

Shapiro-Wilk normality test and Levene tests for the homogeneity of variances were applied to data for the carrageenan content, cystocarp diameter, and number of carpospores released. When data did not satisfy one-way analysis of variance (ANOVA) requirements, non-parametric Kruskal-Wallis test were used. Significant differences between groups were determined by ANOVA followed by a Tukey Unequal N multiple comparisons test or by the non-parametric Dwass-Steel-Critchlow-Fligner test. Statistical significance thresholds were defined as follows: $p < 0.01$ (*) and $p < 0.001$ (**), with annotations directly marked on tables and figures.

Carrageenan content of morphotypes results is expressed as mean ± standard error of the mean from at least three independents replicates, data were not transformed and SPSS V24.0 software was used for statistical analysis. For variation of cystocarp diameter and carpospore number released, data are expressed as mean ± standard error of the mean from at least four independents replicates of variation of the cystocarp diameter and from three independents replicates of the variation of carpospore number released, both data were not transformed and Jamovi 2.6.44 software was used for statistical analysis. Graphs obtained were made using Jamovi 2.6.44 and R 4.5.2 software.

## Results

### Molecular analysis

Partial sequences of the *cox*1 (540 bp), *rbc*L (470 bp), and RuBisCo spacer (300 bp) regions were obtained from 29 samples collected in Yucatán (S1 Table in S1 File). Three morphotypes, 1, 2, and 3, hereafter referred to as M1, M2, and M3, respectively, were identified. Based on this sequence data, the molecular analysis indicates that the specimens collected in this work, regardless of morphotype, were not significantly different from *E. isiformis* (Fig 1).

Our *E. isiformis* samples showed 100% sequence similarity with previously published *rbc*L sequences (GenBank accessions MG948344 and AF099691) from Yucatán, Mexico [32], and Florida, USA [66]. Likewise, *cox*1 sequences (MG948345 and MN941986) from Campeche, Mexico [32], and Florida, USA [67] also showed 99% identity with our samples. The genetic divergence between *E. isiformis* from Yucatán [32] and the morphotype individuals analyzed in this study was 0.80% for the *rbc*L, 0.99% for *cox*1 and RuBisCo region. In contrast, the divergence between our morphotypes and *Eucheuma denticulatum* was 7% for *rbc*L gene. No genetic differentiation was observed among morphotypes based on molecular markers. Among the 29 samples collected in Yucatán, three haplotypes were detected, differing by only a single nucleotide (S1 Table in S1 File). The reference sequence of *E. isiformis* (GenBank MG948361) differed by four nucleotides from Haplotype 1 of *E. isiformis* collected in Dzilam.

The phylogenetic tree revealed two main clades: one comprising *Eucheuma–Eucheumatopsis* species, and the other corresponding to *Kappaphycus alvarezii*. Within the *Eucheuma–Eucheumatopsis* clade, two subclades were resolved: one including all sequences previously recorded as *Eucheumatopsis* spp., and the other including the Dzilam samples

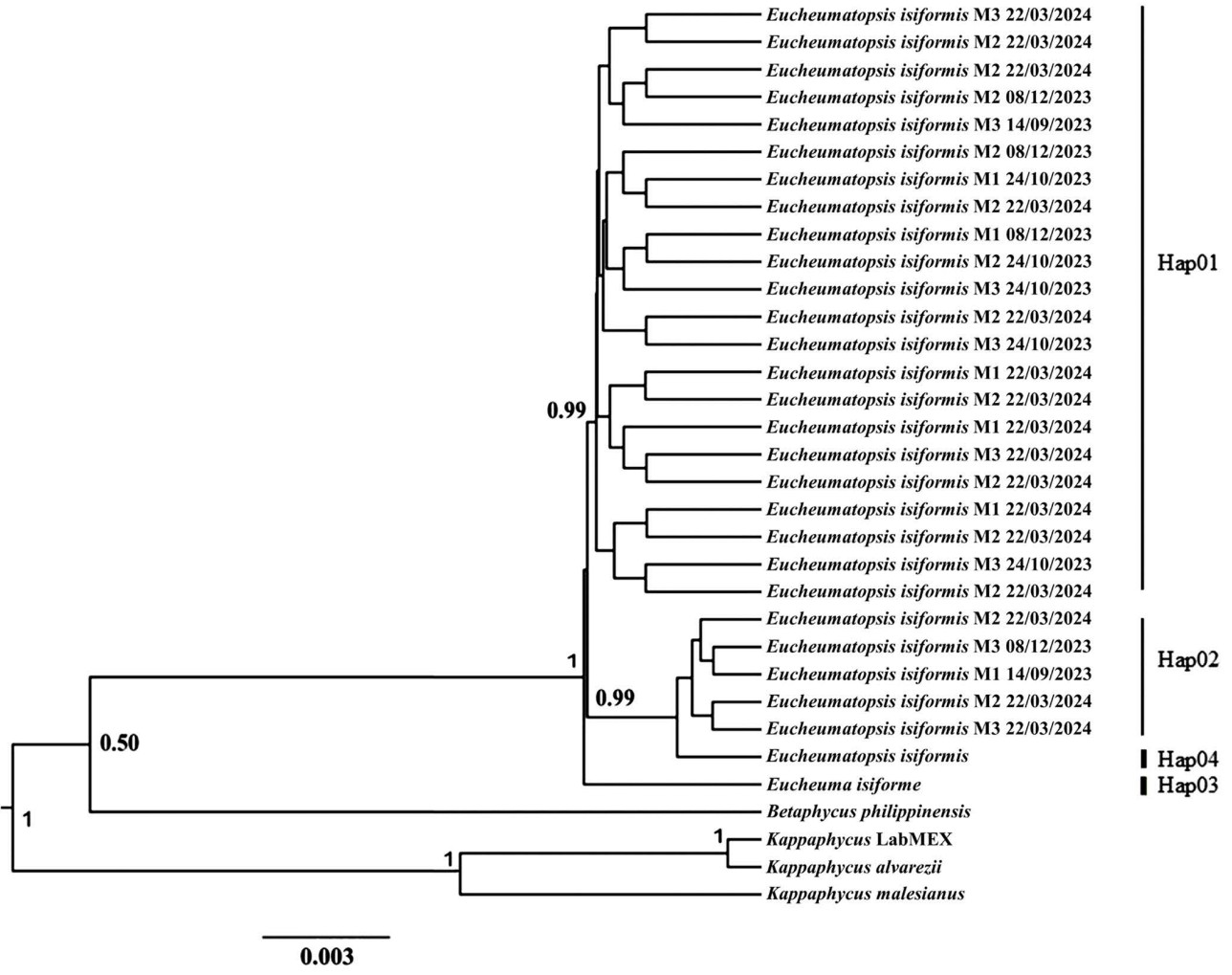

**Fig 1. Bayesian phylogram constructed from concatenated *cox*1, *rbc*L and RuBisCo sequences of *Eucheumatopsis* samples.** Support values are shown only for the main branches and are expressed as Bayesian posterior probabilities.

grouped with *E. isiformis*. No substantial differences were detected in the *cox*1 region sequences among morphotype specimens (M1, M2, M3) collected during different sampling periods.

## Carrageenan content

The carrageenan yield (% dry weight) and carrageenan sulfate content in *E. isiformis* morphotypes collected from March to November 2022 are shown in S2 Table in S1 File. Native carrageenan yield ranged from $39.3 \pm 1.3$ to $58.0 \pm 1.2\%$ dry weight, without significant differences between morphotypes ($F = 0.056$, $p = 0.946$). After alkaline treatment, a 12.6% average yield reduction was observed, concurrent with a 5.8% decrease in sulfate content. The alkaline-treated carrageenan yield ranged from $35.3 \pm 1.1$ to $46.3 \pm 0.8\%$ dry weight. Significant differences were observed between M1 and M2 ($F = 9.57$, $p < 0.01$). Carrageenan yield data met the assumptions of normality ($W = 0.96$, $p = 0.82$) and homogeneity ($F = 0.49$, $p = 0.64$). Analysis of carrageenan yield variation from March to November 2022 revealed that M1 exhibited the maximum yield in July ($57.4 \pm 0.9\%$ dry weight), and it was present during the entire monitoring period in 2022. In the

case of M3, it had an average maximum yield of 58.0 ± 1.2% dry weight in July, whereas M2, showed a maximum yield of 53.6 ± 1.8% dry weight in May, nevertheless it was very scarce during the rest of the year.

## Morphological and anatomical characterization

Morphological and anatomical characteristics of *E. isiformis* morphotypes collected in Dzilam de Bravo are summarized in S3 Table in S1 File. The results revealed contrasting forms, thicknesses, and ramification patterns, although some similarities were also evident. For instance, M2 shared certain morphological features with M1 but differed in the number of lateral branches. In contrast, M3 differed from M1 and M2 in axis and branch thickness and in the presence of spines covering the entire thallus, whereas M1 and M2 had smooth axes and branches in basal areas without spines, and thin lateral branches (Fig 2 A–C). Additional morphological and anatomical differences were observed when these morphotypes were compared with specimens from Florida (S3 Table, S4 Fig in S1 File).

## Reproductive phases

Quantitative data on the reproductive characteristics of *E. isiformis* specimens in the carposporophyte and tetrasporophyte phases, collected between May 2022 and June 2024, are summarized in Table 1.

Specimens bearing cystocarps were present throughout most of the year, with cystocarp diameters ranging from 0.27 to 4.33 mm and pericarp thickness between 34.87 to 208.17 μm in. Cystocarps were predominantly distributed in distal and apical regions, usually occurring singly, though occasionally up to three developed together on a common stalk. They were found on the base, apex, near the apex of short laterals, or along the margins of smooth axes (Fig 2).

In their initial stages of growth, cystocarps are ovoid with a constricted base. As development progresses, they become more ellipsoidal, and a distinct ostiole becomes evident, protruding as the cystocarp matures. Mature cystocarps, measuring from 1.07 to no more than 2.5 mm in diameter, consist of a central fusion cell surrounded by clusters of candelabra-like diploid gonimoblasts bearing carposporangia at the periphery. Filaments extend from these structures into the surrounding pericarp, where they connect with inner medullary cells (Fig 3).

Carposporangia ranged from 2.76 to 50.06 μm in length and from 2.08 to 17.18 μm in width. However, mature carposporangia measured 24.26 to 50.06 μm in length and 6.71 to 17.18 μm in width. Ovoid to spherical carpospores were 14.14–21.33 μm in diameter (Fig 3). Cystocarp diameter differed significantly among months (Table 1) and morphotypes ($X^2 = 144$, $p < 0.001$, Fig 4). Cystocarp diameter data did not meet the assumptions of normality ($W = 0.98$, $p < 0.001$) and homogeneity ($F = 62.5$, $p < 0.001$).

Female carposporophytes bearing cystocarps were present throughout the year, from March 2022 to March 2023 (Fig 4), cartilaginous in texture and exhibited a wide range of red color, as well as yellow and orange, often with a combination of two colors in the same specimen. Distal and proximal axes were terete, typically constricted progressively towards the bases and gradually tapering distal, with smooth axes in the basal areas. Some specimens were completely smooth, whereas others exhibited abundant spines and curved branchlets in distal areas, or had few branchlets and spines concentrated in distal areas (Fig 2 A–C). Thallus length ranged from 1 to 51.2 cm, and axis width ranged from 0.46 to 5.37 mm. Main axes and branches display sympodial, radially alternate, subopposite or irregular branching, typically from the third to fifth order. Apices were acute to rounded, with some forked or truncate, often exhibiting regrowth forming thickened collars at the base. Thalli were multiaxial, with filiform medulla composed of a mass of thick filaments running parallel to the main axes and a pseudoparenchymatous cortex (Fig 3 A–B).

Cystocarp density per 10 cm of thallus ranged from 5 to 19, with the highest values observed in March and the lowest in September and November (Table 1). In March 2023, ostiole differentiation showed that 61.71% were distinguishable as open or opening, 22.95% were in formation but pigmented and closed, and 8.19% were protuberant. Only 7.15% of cystocarps were deformed or broken.

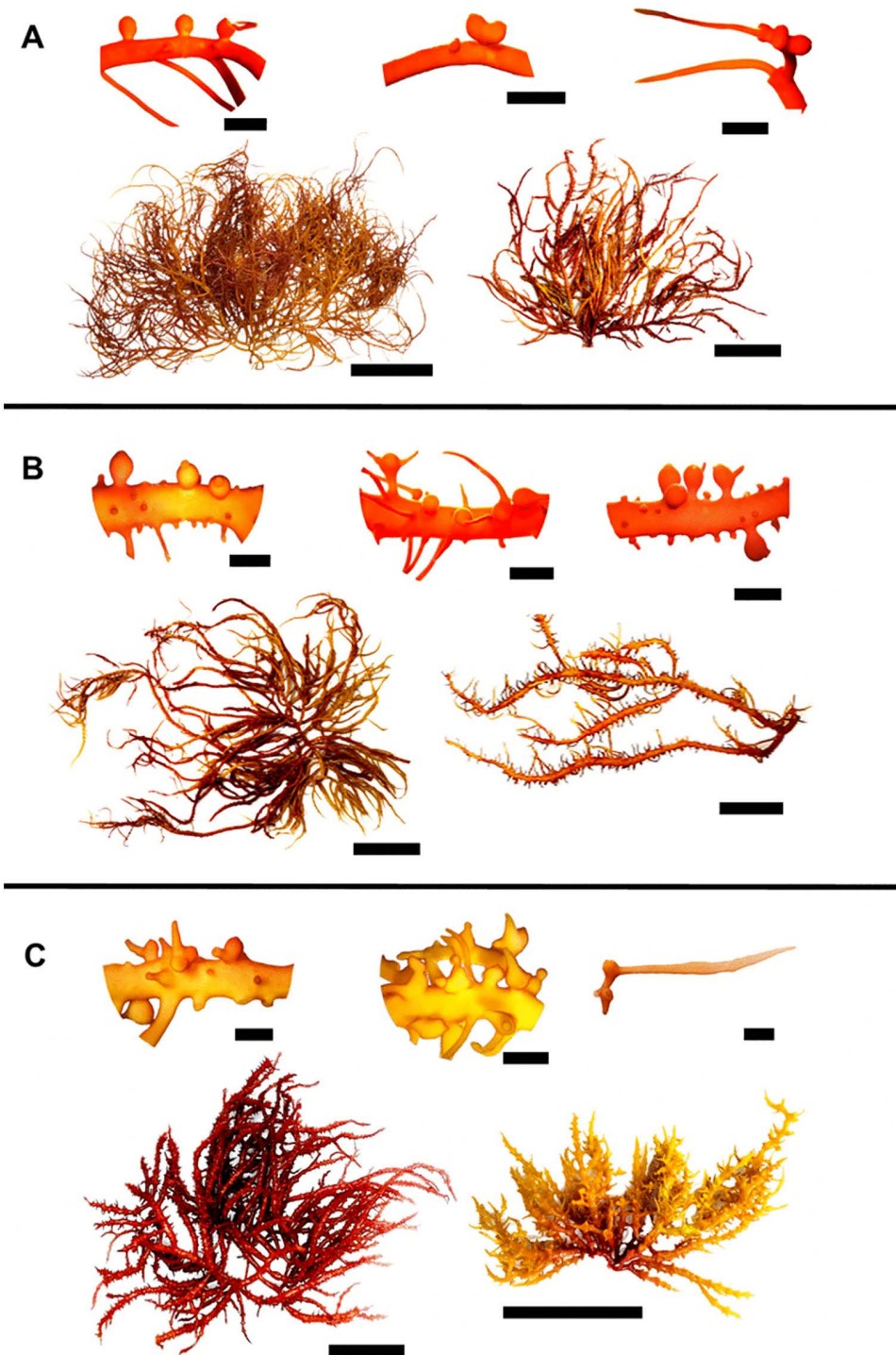

**Fig 2. Morphotype 1 (M1), 2 (M2) and 3 (M3) of *E. isiformis* from Yucatan and their cystocarp diversity. (A)** M1; **(B)** M2; **(C)** M3. Scale bar: complete specimens = 10 cm; cystocarps = 5 mm.

**Table 1. Quantitative data on the reproductive characters of *Eucheumatopsis isiformis* from Dzilam de Bravo, Yucatan.**

| | 2022 | | | 2023 | 2024 |
|---|---|---|---|---|---|
| | **May** | **September** | **November** | **March** | **June** |
| Number of specimens | 13 | 9 | 4 | 10 | 14 |
| Cystocarp density | 8±0.94 (n=54) | 5±0.32 (n=273) | 5±0.83 (n=65) | 29±3.79 (n=30) | – |
| Cystocarp diameter (mm)** | 1.85±0.04 (n=102) | 1.71±0.05 (n=198) | 1.88±0.04 (n=189) | 1.64±0.02 (n=376) | – |
| Pericarp thickness ($\mu$m) | 119±4.20 (n=66) | 122±5.18 (n=30) | 122±7.82 (n=29) | 148±7.15 (n=62) | – |
| Carposporangia length ($\mu$m) | 9.35±0.21 (n=97) | 8.94±0.25 (n=74) | 10.37±0.29 (n=87) | 35.29±0.87 (n=48) | – |
| Carposporangia width ($\mu$m) | 4.39±0.09 (n=97) | 4.82±0.11 (n=74) | 5.05±0.09 (n=87) | 12.25±0.33 (n=48) | – |
| Carpospore diameter ($\mu$m) | – | – | – | 18.1±1.42 (n=5) | – |
| Tetrasporangia length ($\mu$m) | – | – | – | – | 82.3±2.41 (n=61) |
| Tetrasporangia width[b] ($\mu$m) | – | – | – | – | 19.20±0.61 (n=61) |
| Thalli and branch length[b] (cm) | 8.50±0.85 (n=54) | 11.24±0.41 (n=273) | 11.01±0.64 (n=65) | 15.34±1.18 (n=30) | 24.51±2.11 (n=30) |
| Axes and branch diameter (mm) | 3.46±0.11 (n=54) | 2.98±0.09 (n=114) | 3.02±0.11 (n=83) | 2.79±0.09 (n=270) | 4.45±0.35 (n=42) |

** indicates a statistically significant difference between months ($p < 0.001$).

- not data available.

Specimens in the tetrasporophyte phase were observed in 16 out of 24 collected specimens, with 14 specimens collected on June 2023 and August 2024. Tetrasporophyte thalli were cartilaginous in texture, ranging in color from orange to dark red, often displaying singular, more pigmented or discolored spots and lumps resembling the initial growth of branches or ramules (Fig 5). The axes were similar to those of specimens in the carposporophyte phase. Thallus length ranged from 1.79 to 75 cm, and axis width from 1.1 to 12.14 mm.

Tetrasporophyte specimens exhibited second to fourth order branching, and their axes, apices, and overall thallus morphology were similar to those observed in specimens in the carposporophyte phase. However, all tetrasporophyte specimens displayed protuberances across the entire surface of the thallus and branches, as well as small whitish or more pigmented spots or a generally rough texture (Fig 5). Tetrasporangia measured 40.83 to 117.66 µm in length and 9.85 to 29.97 µm in width in June 2024, and 35.11 to 75.61 µm in length and 9.76 to 23 µm in width in August 2024. They were typically elongated-oval to elongated-tubular, with three visible divisions forming a zonate tetrasporangium. Unequal-sized tetraspores were observed, although divisions remained successively zonate. In general, tetrasporangia had elongated ends, with reduced thickness at the oval tips, some showed marked constrictions before the terminal tips (Fig 6).

## Sporulation experiments

Induction of sporulation yielded better results with the application of exogenous polyamines (PUT and SPM) compared with the control at 23ºC. No positive results were obtained at 25°C, as cystocarps began to degrade, specimens lost their cystocarps, or died before the acclimatization period ended. Although the highest number of carpospores was recorded after 5 and 10 days of acclimatization in specimens collected in March 2024 and in February and March 2025, improved

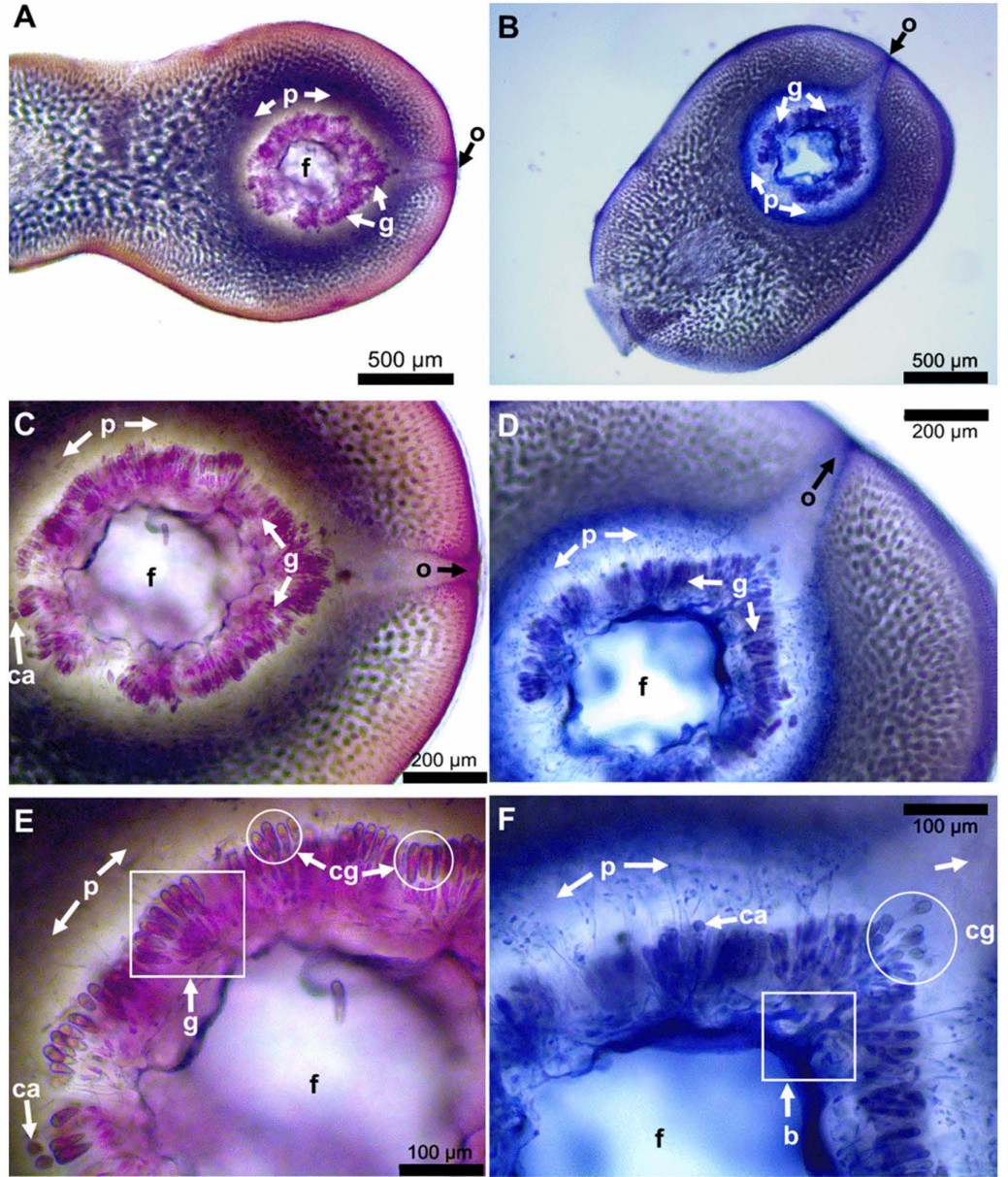

**Fig 3. Mature cystocarps collected in March 2023.** Panels A, C, and E correspond to a cystocarp with diameter of 1.65 mm; whereas panels B, D, and F correspond to a cystocarp with diameter of 1.76 mm. Labels: o = ostiole, f = central fusion cell, p = pericarp: g = gonimoblast packs, cg = carpospo-rangia, ca = carpospores and b = union base of gonimoblast packs arranged in candelabra-like structure.

carpospore development was particularly evident in specimens induced with $10^{-3}$ M putrescine (PUT) after 10 days of acclimatization (Fig 7).

After the 10 days of acclimatization and five days of experimental treatment, the highest number of total carpospores in 5 mL were obtained with $10^{-3}$ M PUT between the first to second day of experiment (Fig 7). Between 72 and 94% of the total number of carpospores were released with $10^{-3}$ M PUT, between 64−85% with $10^{-6}$ M PUT, and between 56–79% for controls of the total carpospores released (Fig 8). There were no significant differences between treatments ($X^2 = 6.36$,

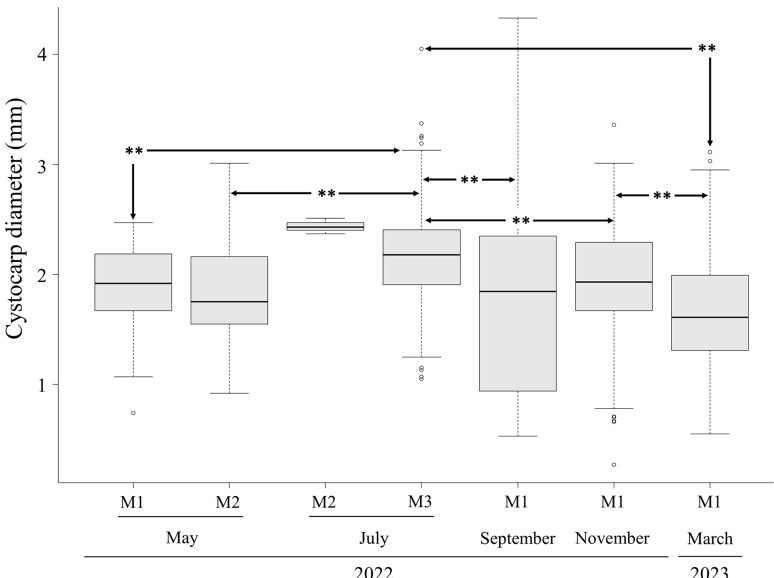

**Fig 4. Variation in cystocarp diameter among morphotypes (M1, M2 and M3) observed from March 2022 to March 2023.** Boxplots represent, from bottom to top, the first quartile, median, third quartile and maximum values, including any outliers. **indicates a statistically significant difference ($p < 0.001$).

$p = 0.61$, Fig 8). However, significant differences were obtained between days ($X^2 = 34$, $p < 0.001$, Fig 9). Carpospore number released data did not meet the assumptions of normality ($W = 0.56$, $p < 0.001$) and homogeneity ($F = 11.4$, $p < 0.001$).

## Discussion

Previous studies on *E. isiformis* [34,37–40] have highlighted its high phenotypic plasticity, suggesting that further investigations are required to clarify the taxonomical status of this taxon. In this study, three morphotypes of *E. isiformis* were recognized (M1, M2, and M3), although an overlap was observed between M1 and M2. The morphotypes found in Yucatán, corresponding to *Eucheuma isiforme* from the Caribbean and the southern Gulf of Mexico, are consistent with previous descriptions [34]. Accordingly, M1 appears to correspond to previous descriptions of *Eucheuma isiforme* var. *denudatum* (with *Eucheuma nudum* as an uncertain record), M2 to the description of *Eucheuma* Bahia Honda form, and M3 to descriptions of *Eucheuma isiforme* var. *isiformis*. However, molecular analysis indicates that all specimens collected in this study, regardless of morphotype, correspond to *E. isiformis* (S2 Table in S1 File). The genetic uniformity among *E. isiformis* morphotypes suggests that their physical variation stems from phenotypic plasticity [68] rather than cryptic speciation [69]. This indicates that morphological traits in this species are plastic responses to environmental heterogeneity, consistent with observations in other eucheumatoids [70].

*E. isiformis* found in coastal areas of the Yucatán Peninsula shows great potential for aquaculture due to its high iota-carrageenan content, rapid growth rates [8–11] and the favorable oceanographic characteristics of the region. Analyses of carrageenan yield revealed that the three morphotypes exhibited similar performance, with values falling in the range previously reported for this species [8,9,71,66]. Additionally, the high sulfate content of the native carrageenan is consistent with values reported for iota-carrageenan-producing *Eucheuma* species [67], suggesting the production of a high-viscosity carrageenan capable of forming transparent and elastic gels [72]. Following alkaline treatment, both carrageenan yield and sulfate groups content decreased in most samples across the three morphotypes. This reduction is likely attributable to the aggressive nature of the alkaline treatment and the heat applied in the processing, which can lead

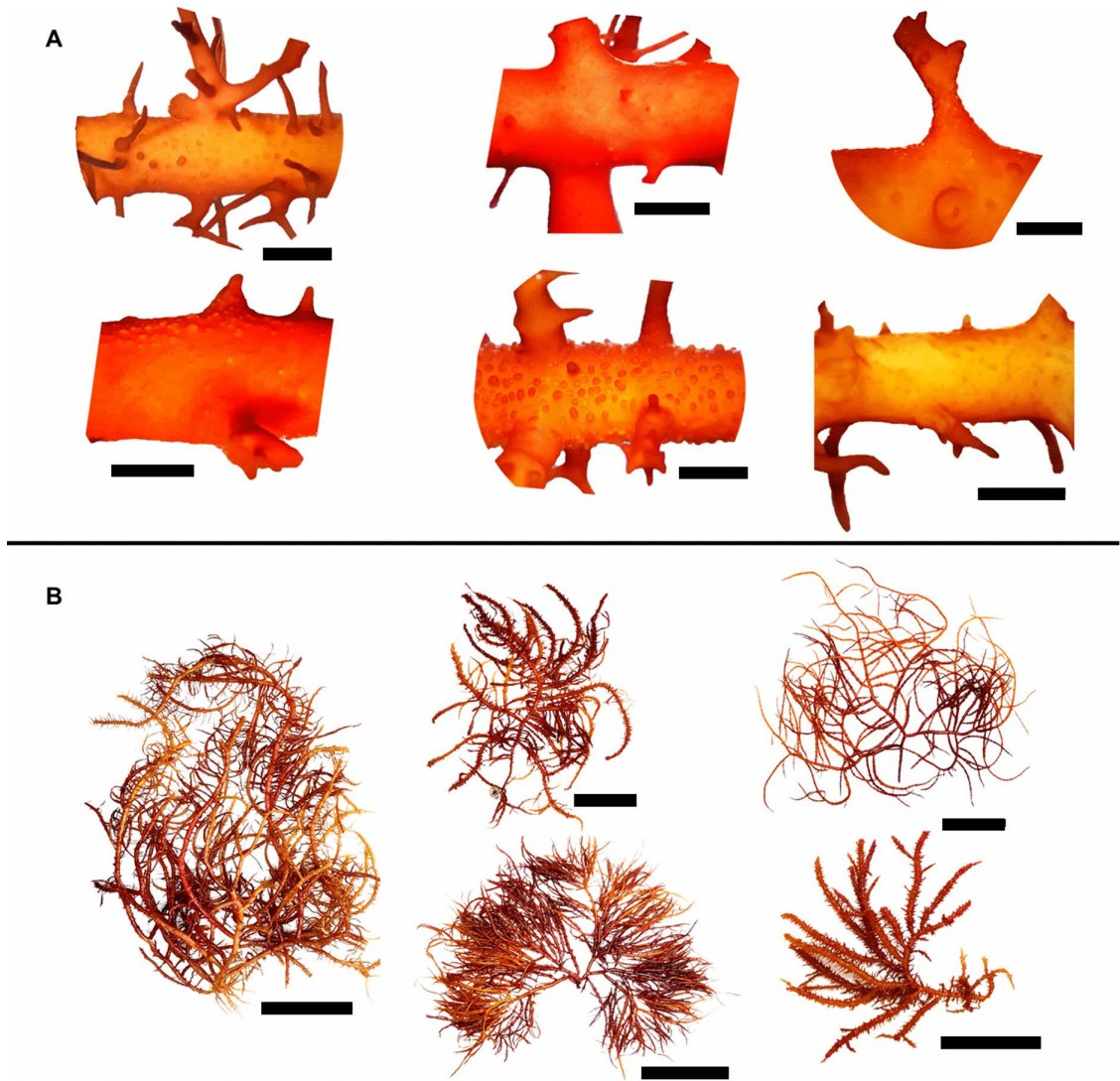

**Fig 5. Morphology of specimens in the tetrasporophyte phase of *E. isiformis* collected in June and August 2024.** Scale bar of sections = 5 mm and complete specimens = 10 cm.

to polysaccharide degradation [73]. In this study, morphotype 1 exhibited a smaller reduction in average yield and sulfate content (8.3% and 3.8%, respectively) compared to morphotypes 2 and 3, suggesting that morphotype 1 may produce carrageenan with superior functional properties. Analysis of temporal variation in carrageenan yield revealed that morphotype 1 and 3 reached their maximum yields in July, whereas morphotype 2 peaked in May, coinciding with the onset of the rainy season [8,74]. This seasonal increase in carrageenan production may be associated with reduced incident light and increased nutrient input to the study area from freshwater sources [8,9]. Seasonal environmental conditions can strongly influence the biochemical composition of carrageenan in E. isiformis by affecting metabolic allocation and cell wall biosynthesis. Reduced light availability during the rainy season may shift carbon allocation from growth toward storage and structural polysaccharides, including carrageenans. Concurrently, increased nutrient input from freshwater runoff, particularly nitrogen and phosphorus, can enhance photosynthetic efficiency and promote polysaccharide synthesis.

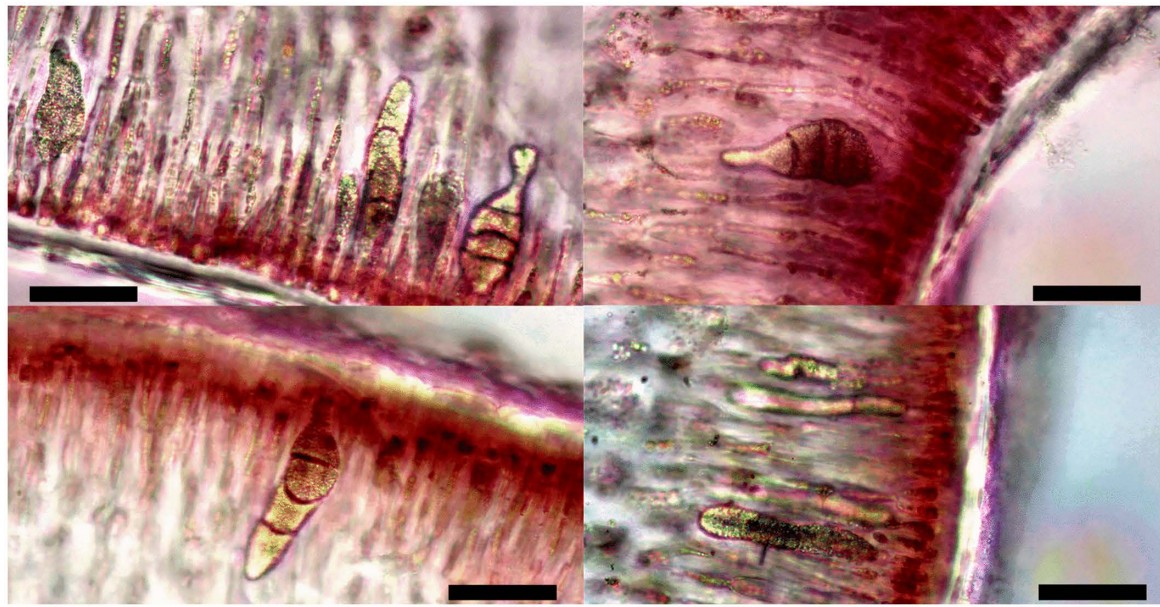

**Fig 6. Tetrasporangia of *E. isiformis* observed in June and August 2024 collections.** Scale bar = 20 μm.

Temperature fluctuations may further modulate enzymatic activity involved in sulfate esterification, influencing the degree of sulfation of carrageenan. Together, these factors likely contribute to the observed seasonal variation in carrageenan yield and sulfate content among morphotypes, with peak production occurring during periods of lower irradiance and higher nutrient availability.

Based on the analysis of morphological and anatomical characters of *E. isiformis* specimens in the carposporophyte phase (Table 1 and Fig 4), it was possible to identify the diagnostic features of fully mature cystocarps (Fig 3), as well as to determine the most suitable months for inducing sporulation and the acclimatization period required to obtain successful results (Fig 7 and 8). In addition, the tetrasporophyte phase was observed only in June and August 2024. Notably, the length and width of tetrasporangia recorded in this study were greater than those reported in previous observations from June 2024 (Table 1). In the case of carposporangia, the maximum length and width recorded in this study were similar to those reported for *Eucheuma serra*, which exhibit mature carposporangia measuring 25–45 μm in length and 10–15 μm in width, [75]. In contrast, *E. isiforme* showed smaller carposporangia, with lengths ranging from 15 to 26 μm and widths from 9 to 14 μm.

Regarding cystocarp density in *E. isiformis*, we recorded between 5–19 cystocarps per 10 cm of thallus, values comparable to those reported for *Eucheuma perplexum*, which exhibits 1–3 cystocarps per 0.1–2 cm [76]. This pattern may be related to the seasonal growth dynamics of *E. isiformis,* which shows active growth from the late cold season (October–January), through the dry season (February–April), when seawater temperature remains optimal (<30°C). Seawater temperatures begin to rise in May, reaching their highest values between July and September [8,74], potentially influencing reproductive output and cystocarp development. Based on historical seawater temperature records [74], observations of cystocarp maturation, and induction tests conducted in June 2023 and from April to August 2024, we confirmed that specimens had already sporulated. Our results indicate that the optimal period for inducing sporulation in the carposporophyte phase of *E. isiformis* occurs when seawater temperature begins to rise slightly and stabilize, between the late February and mid-March, at temperatures not exceeding 24.5°C.

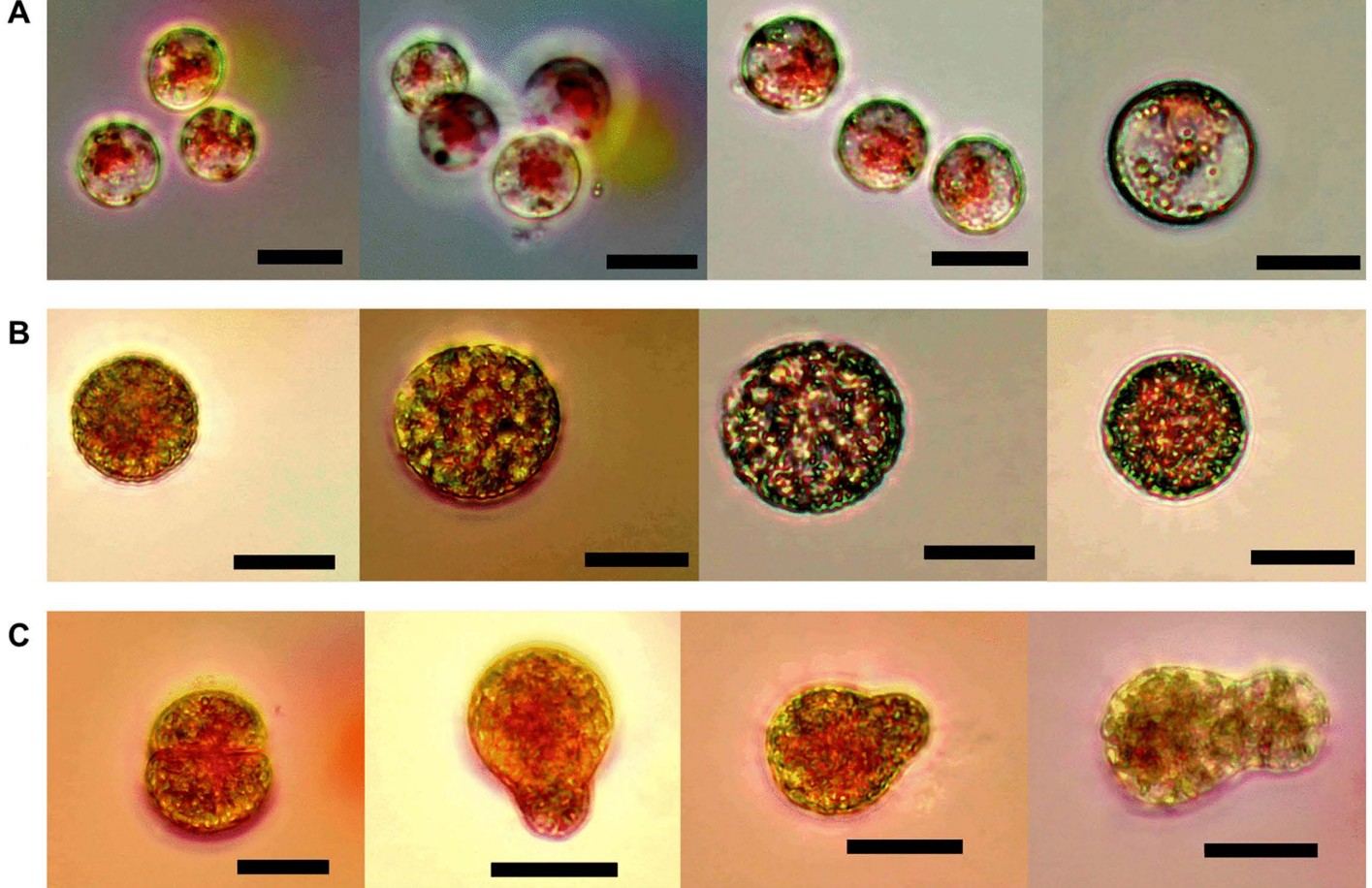

**Fig 7. Carpospores obtained under treatment with 10⁻³ M putrescine (PUT) in March 2024 after 10 days of acclimatization.** (A) dark after 36-37 h of induction. (B) 20 μmol·m⁻²·s⁻¹ after 38 h of induction. (C) 40 μmol·m⁻²·s⁻¹ after 34-38 h of induction. Scale bar = 20 μm.

Notably, from July to November 2023, no specimens were observed in the study area, and from April to August 2024, only 2–4 specimens in the carposporophyte phase with fully mature cystocarp were found. These deviations from collections in 2022 may be related to recent periods of elevated ocean temperature, which reached historical high levels over the past two years [74], potentially affecting reproductive timing and population abundance.

As has been shown for other red tropical seaweed, the combined effects of moderate temperature [77–79], reduced light stress [80], and increased nutrient availability [81] likely synchronize reproductive events in *E. isiformis*, explaining the observed seasonal patterns in cystocarp density, sporulation timing, and the restricted occurrence of tetrasporophytes.

Previous studies [38,39,62–65] were essential for designing the experimental conditions to induce sporulation in *E. isiformis*, particularly regarding environmental factors at the collection site and key drivers of sporulation such as temperature, light intensity, and photoperiod. For related species from Florida, optimal light requirements are relatively low, with light saturation occurring between 60 and 180 μmol·m⁻²·s⁻¹. Similarly, the optimal temperature range is 21–24°C, which is comparatively low relative to the seasonal temperatures these species typically experience. For the Bahia Honda form of *E. isiformis* (M1), reported optimal temperatures are 23 and 24°C [63,82].

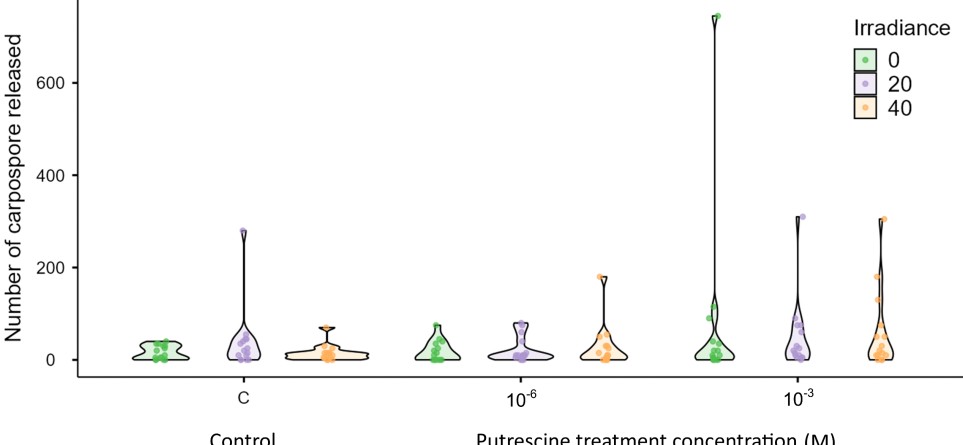

**Fig 8. Number of carpospores obtained in March 2024 after 10 days of acclimatization.** Irradiances levels are indicated as 0, 20 and 40 µmol·m⁻²·s⁻¹. Treatments included two Putrescine concentrations $10^{-3}$ and $10^{-6}$ M, and control (C). Error bars represent standard error of the mean (SEM) for three independent replicates.

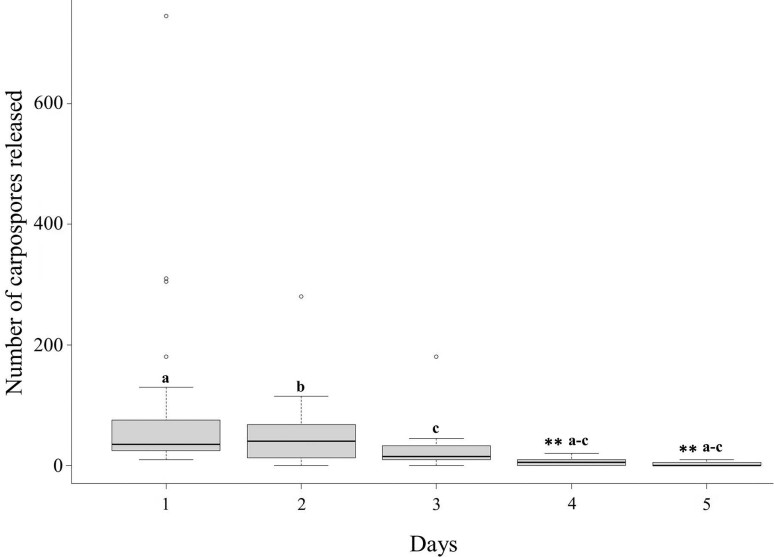

**Fig 9. Number of carpospores released per day (1-5) in March 2024 after 10 days of acclimatization.** Boxplots represent, from bottom to top, the first quartile, median, third quartile and the maximum values, including outliers.**indicates a statistically significant difference ($p < 0.001$).

Previous studies successfully induced sporulation in different red seaweed species using exogenous inducers [41,42], but not in *E. isiformis*. For example, in *Gracilaria cornea,* favorable conditions for carpospore release occurred under low irradiance, with optimal results at 26°C, 40 µmol·m⁻²·s⁻¹ and 8:16 h L:D photoperiod [65], by comparing different concentrations and combinations of exogenous inducers, including putrescine (PUT), spermine (SPM) and spermidine (SPD), this study found that $10^{-3}$ M PUT significantly enhanced carpospore release in all treatments. Our results in *E. isiformis* indicate that by using $10^{-3}$ M PUT we obtained significantly enhanced carpospore release in all treatments, after approximately 5 days of acclimatization, with 10 days ideal for optimal carpospore development. Induction efficiency could also

be enhanced with 7–8 days of acclimatization, particularly when seawater temperature begins to rise and stabilize, as occurs in this tropical region between late February and mid-March. Moreover, morphological differences were observed under different irradiance levels, including faster carpospore development. This represents the first successful induction of sporulation in *E. isiformis*, providing a methodological basis for cultivation from spores and highlighting the need to optimize conditions for establishing plantlets. Future research should explore phylogenetic relationships among western Atlantic populations and further evaluate the tetrasporophyte phase. Wild morphotypes from the Caribbean Sea and southern Gulf of Mexico could serve as a seedbank for scalable culture and domestication efforts.

## Conclusion

This study confirms the identity of *E. isiformis* for the three described morphotypes, which exhibited similar carrageenan yield. For future spore induction of *Eucheuma* and *Eucheumatopsis* species, we recommend recording quantitative data on mature carposporangia and tetrasporangia, monitoring ostiole characteristics during the cystocarp maturation, and, when possible, tracking physiological changes during acclimatization for each reproductive phase to improve carpospore development. Overall, these findings provide a methodological foundation for selective breeding and domestication protocols for *E. isiformis* and offer a framework that could be applied to other eucheumatoid species to support aquaculture development.

## Supporting information

**S1 File. S1 Table.** Molecular analysis. Coding, date of collection and haplotype description used for construction of the Bayesian phylogram of Eucheumatopsis from Yucatán, Mexico. Haplotype description is based on gene sequencing of cox1, rbcL, and RuBisCo spacer. **S2 Table.** Carrageenan content. Native and alkali-treated carrageenan yields and sulfate content of Eucheumatopsis isiformis for the different morphotypes collected from March to November 2022. **S3 Table.** Morphological and anatomical characterization. Morphological characteristics of Eucheumatopsis isiformis morphotypes. Comparisons with specimen from Florida, United States of America (USA). **S4 Fig.** Morphological characterization. Specimen identified as Eucheumatopsis isiformis from Bahia Honda, Florida, USA.
(ZIP)

## Acknowledgments

Doctoral studies of Monserrat López Yllescas (CVU 508791) were supported by the National Council of Humanities, Sciences, and Technologies of Mexico (scholarship 2022-000002-01NACF-05280). We thank Román Vásquez Elizondo and Víctor Ávila Velázquez for their support during fieldwork. Crescencia Chávez Quintal (Cinvestav), Mayra Sánchez-García (MBL), Ana Isabel González Luna, Javier Robles Flores, and Paola Saritzia Ruiz Tamayo (UABC) provided invaluable technical support and expertise during laboratory and molecular analyses. We are also grateful to the anonymous peer reviewers and the public peer reviewer, Rajasekar Thirunavukkarasu, for their substantial contributions in improving our manuscript.

## Author contributions

**Conceptualization:** Monserrat López-Yllescas, Daniel Robledo.

**Data curation:** Monserrat López-Yllescas, Erika F. Vázquez-Delfín, Adrián Fagundo-Mollineda, Yolanda Freile-Pelegrín, Raquel Muñiz-Salazar, Daniel Robledo.

**Formal analysis:** Monserrat López-Yllescas, Erika F. Vázquez-Delfín, Adrián Fagundo-Mollineda, Raquel Muñiz-Salazar.

**Funding acquisition:** Loretta M. Roberson, Daniel Robledo.

**Investigation:** Monserrat López-Yllescas, Yolanda Freile-Pelegrín, Daniel Robledo.

**Methodology:** Monserrat López-Yllescas, Erika F. Vázquez-Delfín, Adrián Fagundo-Mollineda, Yolanda Freile-Pelegrín, Raquel Muñiz-Salazar, Daniel Robledo.

**Project administration:** Loretta M. Roberson, Daniel Robledo.

**Resources:** Erika F. Vázquez-Delfín, Adrián Fagundo-Mollineda, Yolanda Freile-Pelegrín, Raquel Muñiz-Salazar, Loretta M. Roberson, Daniel Robledo.

**Supervision:** Loretta M. Roberson, Daniel Robledo.

**Validation:** Erika F. Vázquez-Delfín, Yolanda Freile-Pelegrín, Raquel Muñiz-Salazar, Loretta M. Roberson.

**Visualization:** Monserrat López-Yllescas, Erika F. Vázquez-Delfín, Adrián Fagundo-Mollineda, Raquel Muñiz-Salazar, Daniel Robledo.

**Writing – original draft:** Monserrat López-Yllescas, Erika F. Vázquez-Delfín, Adrián Fagundo-Mollineda, Yolanda Freile-Pelegrín, Raquel Muñiz-Salazar, Daniel Robledo.

**Writing – review & editing:** Monserrat López-Yllescas, Erika F. Vázquez-Delfín, Yolanda Freile-Pelegrín, Loretta M. Roberson, Daniel Robledo.

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
