## [Decision Letter · Decision Letter 0]

10 Nov 2025

Dear Dr. Daniel Robledo,

Thank you for submitting your manuscript to PLOS ONE. After careful consideration, we feel that it has merit but does not fully meet PLOS ONE’s publication criteria as it currently stands. Therefore, we invite you to submit a revised version of the manuscript that addresses the points raised during the review process.

The reviewers recognize the scientific merit and relevance of your work, which provides important insights into the reproductive biology, biochemical composition, and cultivation potential of a tropical carrageenophyte.

After careful evaluation, the editorial board has concluded that your manuscript requires major revision before it can be considered for publication. The reviewers have provided constructive comments focusing on statistical clarity, consistency in data presentation, and overall structural refinement to enhance readability and scientific impact.

We encourage you to carefully address all reviewer suggestions and resubmit a thoroughly revised version, highlighting the changes made and providing a detailed response to each comment.

We appreciate your contribution to the field and look forward to receiving your revised manuscript.

We look forward to receiving your revised manuscript.

Kind regards,

Inbakandan Dhinakarasamy, Ph.D

Academic Editor

PLOS ONE

“This study was funded by the project ARPA-e DE-AR0000912 - Sub-award 54336 Development of Techniques for the Cultivation of Tropical Algae.”

4. Thank you for stating the following in the Funding Section of your manuscript:

“This study was funded by the project ARPA-e DE-AR0000912 - Sub-award 54336 Development of Techniques for the Cultivation of Tropical Algae.”

“This study was funded by the project ARPA-e DE-AR0000912 - Sub-award 54336 Development of Techniques for the Cultivation of Tropical Algae.”

Additional Editor Comments (if provided):

The manuscript presents valuable baseline information on the reproductive biology, biochemical composition, and cultivation potential of a native tropical carrageenophyte, offering important insights into its role in sustainable aquaculture. The study is scientifically relevant and timely, and it successfully integrates morphological, molecular, and biochemical approaches. However, the manuscript requires substantial revision before it can be considered for publication.

The statistical analysis needs improvement through the inclusion of variability measures such as mean ± SD or SE and clear indication of significance among treatments. The rationale for using non-parametric tests should be clarified by specifying which datasets failed normality and reporting corresponding test results. The organization of the text should be refined with appropriate subheadings in the Methods and Results sections to enhance readability and logical flow. Simplifying long sentences and reducing repetitive numerical details will make the manuscript more concise and engaging. Figures and tables need to consistently indicate the number of replicates, statistical notations, and appropriate salinity units (PSU or ppt), with improved legends that summarize the main findings. Consistency in terminology, species naming, abbreviations, and reference formatting is also necessary.

The introduction should be condensed to focus on the rationale and novelty of the study, highlighting how it differs from previous work on related species. Incorporating recent literature (2023–2024) would strengthen its scientific depth. The discussion could be improved by emphasizing the biological and aquacultural implications of the results, particularly how morphological variation, carrageenan characteristics, and induced sporulation relate to environmental adaptation and cultivation potential.

In summary, the manuscript has strong scientific potential and aligns well with the scope of the journal. With careful attention to structure, statistical rigor, and clarity in interpretation, it could become a valuable contribution to the field.

Reviewers' comments:

Reviewer's Responses to Questions

**Comments to the Author**

1. Is the manuscript technically sound, and do the data support the conclusions?

Reviewer #1: Yes

Reviewer #2: Yes

Reviewer #3: Yes

2. Has the statistical analysis been performed appropriately and rigorously?

Reviewer #1: Yes

Reviewer #2: Yes

Reviewer #3: Yes

3. Have the authors made all data underlying the findings in their manuscript fully available?

Reviewer #1: Yes

Reviewer #2: Yes

Reviewer #3: Yes

4. Is the manuscript presented in an intelligible fashion and written in standard English?

Reviewer #1: Yes

Reviewer #2: Yes

Reviewer #3: Yes

Reviewer #1: This manuscript has potential to publish, and, no doubt that the manuscript and the topic is interesting, significant and fits very well in the core topics covered by the journal. However, my opinion is, there are still major improvements needed to accept this manuscript for publication.

General Comments:

1. Please explain. how do environmental conditions in the season (temperature, light, nutrient availability) influence reproductive phenology and carrageenan biochemical composition of E. isiformis?

2. Do the morphotypes in the collection site adapt to different microhabitats and is possibly the reason of morphological differences but not molecular divergence?

3. How does the change of the carrageenan sulphate content of the morphotypes have any functional value, especially under the conditions of use in the industry?

Specific comments:

Abstract suggest to reduce and emphasize the key findings for clarity.

Line 15 and 16: “biomass fitness” → “biomass decline,” “lack of genetic exchange” → “limited genetic exchange.” Suggested word changes:

Line 38- 49: Repetition of “genetic diversification.” Please check once.

Line 49- 56: Some sentences appear long and complex, which can short sentences to improve clarity.

Proofread the entire paper and maintain abbreviations (e.g., Eucheumatopsis isiformis abbreviated to E. isiformis).

Line 114: The manuscript alternates between referring to the species as Eucheumatopsis isiformis and its synonym Eucheuma isiforme. Consistent usage throughout would avoid confusion.

Line 81 to 211: Detailed technical protocols are included, but use subheadings to break down the methods into separate sections (e.g., DNA extraction, PCR, carrageenan extraction) to improve readability.

Table 3 contains dense data, but lacks descriptive legends that clarify significance.

Sections sometimes jump suddenly from one topic to another (e.g., molecular analysis to carrageenan extraction).

Reviewer #2: I have carefully reviewed the manuscript entitled

“Studies on the tropical carrageenophyte Eucheumatopsis isiformis (Solieriaceae, Rhodophyta) from Yucatán, Mexico: cultivation potential based on morphological, chemical, and reproductive features.” This manuscript presents valuable baseline information on the reproductive biology, biochemical composition, and cultivation potential of the red alga Eucheumatopsis isiformis collected from Yucatán, Mexico. The topic is both relevant and timely, as it contributes to the understanding of native eucheumatoid species and their potential applications in sustainable aquaculture, an area of increasing scientific and economic importance.

The study integrates multiple aspects including field sampling, morpho-anatomical characterization, molecular analysis, and carrageenan composition, offering a comprehensive foundation for further research on E. isiformis.

However, while the study is scientifically significant, the manuscript requires structural refinement, stronger statistical justification, and improved consistency in data presentation to meet publication standards. My detailed comments are summarized below for your consideration.

Major Comments

1. Lack of variability reporting in polyamine-induced sporulation experiment

In the results of the polyamine-induced sporulation experiment (Page 378–382; Fig. 6), only the carpospore counts are presented (e.g., “363 carpospores per 5 mL…”), without reporting standard deviation (SD) values or error bars in the figure. Although the “Materials and Methods” section (Page 10, Lines 195–199) indicates that each treatment was performed in triplicate (n = 3), this information is not reflected in the results. The authors should include mean ± SD (or SE) values and, if applicable, significance indicators among treatments to ensure statistical completeness and data reliability.

2. Unclear justification for the use of non-parametric tests

The manuscript applies both parametric (ANOVA) and non-parametric (Kruskal–Wallis) tests, but the justification for using the non-parametric approach is unclear. Although the Statistical Analysis section (Page 10–11, Lines 208–221) mentions the use of Shapiro–Wilk and Levene’s tests to assess normality and homogeneity, the authors do not specify which dataset(s) failed these assumptions. Please clarify which data did not meet normality, report corresponding test results (e.g., p < 0.05), and explain the rationale for applying non-parametric analysis.

Minor Comments

1) Salinity units:

The unit of salinity should be expressed consistently throughout the manuscript according to the instrument used for measurement. Please use PSU (Practical Salinity Unit) or ppt (parts per thousand) instead of ppm, which is not appropriate for salinity reporting in marine studies.

2) Incomplete references:

Some references are missing publication years (e.g., “Refs. [37–38, 62–66]”). Please check all references carefully and complete missing bibliographic information in accordance with the journal’s reference style.

3) Statistical notation consistency:

Ensure consistent formatting of statistical terms (e.g., n = 3, p < 0.05, mean ± SD) throughout the manuscript, tables, and figure captions.

4) Figure legends:

Include the number of replicates (n) and clarify whether values are presented as mean or median in all figure legends.

5) Abbreviation clarity:

Check that all abbreviations (e.g., SPE, PUT, PES) are defined at first mention in both the text and figure captions.

In summary, the manuscript is scientifically promising and provides useful data for future studies on the reproductive biology and mariculture potential of E. isiformis. However, revisions are necessary to improve clarity, statistical transparency, and some issues that require correction before it can be considered for publication.

Reviewer #3: Studies on the tropical carrageenophyte Eucheumatopsis isiformis (Solieriaceae, Rhodophyta) from Yucatán, Mexico: cultivation potential based on morphological, chemical, and reproductive features”

1. Title is scientifically sound and informative, but it is quite long and descriptive.

May be considered

Cultivation potential of the tropical carrageenophyte Eucheumatopsis isiformis (Solieriaceae, Rhodophyta) from Yucatán, Mexico

2. The abstract effectively highlights the cultivation potential of Eucheumatopsis isiformis; it could be improved by briefly summarizing the key morphological and molecular differences among the analyzed morphotypes to strengthen the connection between diversity and cultivation potential.

3. The introduction is too lengthy, which may overwhelm readers.

4. Some background information especially on general seaweed life cycles and previously studied Eucheuma species—could be condensed to maintain focus on Eucheumatopsis isiformis and the specific research gap.

5. Although the section includes relevant citations up to 2022, it lacks very recent studies (2023–2024) on seaweed genetic improvement, reproductive biology, or carrageenan yield optimization. Including these would enhance the currency and scientific depth of the background.

6. The introduction should more clearly highlight why E. isiformis was chosen, how it differs from commercial Eucheuma species, and what specific research questions or hypotheses this study addresses.

7. The section is well-structured and provides comprehensive methodological details for replication. Provide subheadings with clearer separation (e.g., Sample processing, Microscopy and staining, Data analysis) to enhance readability, as the current format is text-heavy and occasionally dense.

8. The sporulation induction and carrageenan extraction procedures are well explained, but the rationale for selecting specific concentrations of polyamines and irradiance levels could be briefly justified with relevant citations or preliminary data to strengthen methodological reasoning.

9. The section demonstrates appropriate use of statistical tests; however, the choice of parametric vs. non-parametric methods could be better justified. Additionally, information on software used for analysis (e.g., SPSS, R, or GraphPad) should be explicitly mentioned for transparency and reproducibility.

10. Some sentences are overly long and could be simplified. There are minor typographical errors (e.g., double periods in “PCR conditions.” and “residual Cetavlon.”). Consistent use of units (e.g., “μmol·m⁻²·s⁻¹”) and spacing around symbols should be checked for formatting accuracy.

11. The result section provides rich data but could benefit from clearer subheadings for each main result (e.g., Morphoanatomical characterization, Carrageenan content, Molecular analysis, Reproductive phases, Sporulation experiments). This would enhance readability and help the reader follow the logical sequence of findings.

12. The text contains excessive numeric detail (e.g., every mean and standard deviation is listed in sentences as well as in tables). This makes the narrative dense and repetitive. Consider focusing the text on key trends, comparisons, and significant findings, while leaving raw data to tables and supplementary files.

13. The integration of figures and tables is appropriate, but figure legends and table references could be clearer. Some figure legends (e.g., Figs. 6–7) could briefly summarize the key outcome (e.g., “Carpospore release enhanced under 10⁻³ M PUT treatment compared with control”).

14. While some statistical results (e.g., F=9.57, p<0.01; H=75.74, p<0.01) are reported, interpretation of their biological meaning is limited. It would strengthen the result section to discuss what these differences imply e.g., whether morphotype differences correlate with environmental adaptation, or whether the yield reduction post-alkaline treatment is significant for industrial use.

15. The molecular analysis is concise and clear, but lacks a visual reference (e.g., phylogenetic tree figure or haplotype network). The paragraph would be stronger if you explicitly noted whether the lack of genetic differentiation despite morphological variability suggests phenotypic plasticity or cryptic speciation.

16. The experiment is well-detailed, but results and interpretation are somewhat blended with methods. For instance, repeated mentions of acclimatization periods, temperature, and irradiance fit better in the Methods. Here, focus on outcome patterns e.g., “Putrescine (10⁻³ M) significantly enhanced carpospore release at lower irradiance, although variability across days was noted” to clarify the biological relevance.

17. The discussion contains a lot of detailed data and literature references, which demonstrate strong experimental support. However, it would benefit from more concise synthesis clearly highlighting how your findings advance understanding of E. isiformis biology or cultivation compared to previous studies. Some paragraphs could be shortened by summarizing numerical data instead of repeating all values.

18. While the text notes that no genetic differences were observed among morphotypes, it could be strengthened by discussing the implications of this finding for example, whether this morphological variability is due to environmental plasticity or phenotypic adaptation.

19. The discussion ends with good experimental observations on polyamine-induced sporulation, but it should also clearly state the broader implications such as how these findings contribute to large-scale cultivation strategies or seedbank development for tropical carrageenophytes.

20. Adding 2–3 lines on how this work supports future selective breeding or aquaculture resilience would make the conclusion stronger and more impactful.

21. The conclusion clearly outlines future research directions but could be more concise and focused on the main findings such as species identification, morphotype characterization, and successful sporulation induction before discussing future perspectives.

22. It would be beneficial to separate experimental details from general recommendations, presenting a clearer summary of key outcomes followed by suggested future studies for better readability and impact.

.

Reviewer #1: No

Reviewer #2: No

Reviewer #3: **Yes:** Rajasekar ThirunavukkarasuRajasekar ThirunavukkarasuRajasekar ThirunavukkarasuRajasekar Thirunavukkarasu

---

## [Author Response · Author response to Decision Letter 1]

10 Feb 2026

Prof. Inbakandan Dhinakarasamy

Academic Editor PLOS ONE

Enclosed you will find the revised version of the manuscript PONE-D-25-36873R1 including a rebuttal letter that responds to each point raised by you as academic editor and the comments and suggestions by the three reviewers who evaluate the manuscript first version. Moreover, we also included a marked-up copy of the manuscript highlighting all the changes made to the original version. You may see that we have made substantial arrangements in the manuscript section to clarify and specify our main goal and findings. An unmarked version of the revised paper without tracked changes (Clean version) is included to be revised.

In relation to your general comments: (1). We have reviewed PLOS ONE’s style requirements and made revisions where appropriate, including verification of figure configuration through Newgen art analysis. (2). In the Methods section, please provide additional information regarding the permits you obtained for the work. Please ensure you have included the full name of the authority that approved the field site access and, if no permits were required, a brief statement explaining why. We have included a sentence to justify seaweed collection “Collection does not require a license or permit to harvest seaweed by hand for scientific use”.

(3). Thank you for stating the following financial disclosure: “This study was funded by the project ARPA-e DE-AR0000912 - Sub-award 54336 Development of Techniques for the Cultivation of Tropical Algae.” We have updated the role of funder "The funders had no role in study design, data collection and analysis, decision to publish, or preparation of the manuscript."

(4). Thank you for stating the following in the Funding Section of your manuscript:

“This study was funded by the project ARPA-e DE-AR0000912 - Sub-award 54336 Development of Techniques for the Cultivation of Tropical Algae.”

Funding information has been removed from the Acknowledgments section and is now presented in the Funding Statement section of the online submission form.

'Response to Reviewers'.

Additional Editor Comments:

The manuscript presents valuable baseline information on the reproductive biology, biochemical composition, and cultivation potential of a native tropical carrageenophyte, offering important insights into its role in sustainable aquaculture. The study is scientifically relevant and timely, and it successfully integrates morphological, molecular, and biochemical approaches. However, the manuscript requires substantial revision before it can be considered for publication.

1. The statistical analysis needs improvement through the inclusion of variability measures such as mean ± SD or SE and clear indication of significance among treatments. The statistical analysis was improved and clarified across all sections of the manuscript. For example, the mean ± standard error (SE) for carrageenan content and measured reproductive characters are now specified in S2 Table and Table 1. In addition, significant differences between treatments are indicated in Figures 4 and 7 for the number of carpospores released, and are described in the corresponding text.

2. The rationale for using non-parametric tests should be clarified by specifying which datasets failed normality and reporting corresponding test results. The rationale for using nonparametric statistics has been included, particularly for cystocarp diameter variation and the number of released carpospores, which did not meet the assumptions of normality and homogeneity of variance. This information has been added to the corresponding sections, and nonparametric tests were applied for their statistical analysis (Lines 402–408 and 500–502).

3. The organization of the text should be refined with appropriate subheadings in the Methods and Results sections to enhance readability and logical flow. As shown in the track-changes version, substantial modifications were made to improve readability and clarity throughout the manuscript text. The subheadings in the Methods and Results sections were reorganized in a logical and sequential

manner.

4. Simplifying long sentences and reducing repetitive numerical details will make the manuscript more concise and engaging. The manuscript was improved by avoiding long sentences and repetitive numerical details. All sentences were reviewed and revised accordingly. For example, data on reproductive characters were streamlined and are now referenced in Table 1, where values are presented as mean ± standard error of the mean.

5. Figures and tables need to consistently indicate the number of replicates, statistical notations, and appropriate salinity units (PSU or ppt), with improved legends that summarize the main findings. In the Methods and Results sections, the number of replicates, statistical notations, and appropriate salinity units were specified, and all abbreviations (e.g., PSU) were corrected. These changes improve clarity and help to better summarize the main findings of the study.

6. Consistency in terminology, species naming, abbreviations, and reference formatting is also necessary. All species abbreviations throughout the document were reviewed. The abbreviation for Eucheumatopsis isiformis was retained, while species belonging to Eucheuma that are mentioned explicitly were left unabbreviated to avoid confusion. In addition, all citations and references were reviewed, renumbered, and reordered where necessary, and their formatting was checked.

7. The introduction should be condensed to focus on the rationale and novelty of the study, highlighting how it differs from previous work on related species. Incorporating recent literature (2023–2024) would strengthen its scientific depth. The Introduction was shortened by condensing previously presented information on E. isiformis and incorporating relevant recent literature (Line 51, Reference 5). The rationale for selecting this species was clarified, including the characteristics that differentiate it from other commercial species and the knowledge gaps addressed by this study. Greater emphasis was also placed on the need for cultivating this species from spores and the potential impact of this approach on the mariculture of eucheumatoids.

8. The discussion could be improved by emphasizing the biological and aquacultural implications of the results, particularly how morphological variation, carrageenan characteristics, and induced sporulation relate to environmental adaptation and cultivation potential. All suggestions to improve the Discussion and Conclusion were carefully considered. As a result, the manuscript was substantially revised—particularly in these sections—to improve structure, statistical rigor, and clarity in the interpretation of the results.

Reviewer's Responses to Questions

Comments to the Author

1. Is the manuscript technically sound, and do the data support the conclusions?

Reviewer #1: Yes

Reviewer #2: Yes

Reviewer #3: Yes

2. Has the statistical analysis been performed appropriately and rigorously?

Reviewer #1: Yes

Reviewer #2: Yes

Reviewer #3: Yes

3. Have the authors made all data underlying the findings in their manuscript fully available?

Reviewer #1: Yes

Reviewer #2: Yes

Reviewer #3: Yes

4. Is the manuscript presented in an intelligible fashion and written in standard English?

Reviewer #1: Yes

Reviewer #2: Yes

Reviewer #3: Yes

5. Review Comments to the Author

Reviewer #1: This manuscript has potential to publish, and, no doubt that the manuscript and the topic is interesting, significant and fits very well in the core topics covered by the journal. However, my opinion is, there are still major improvements needed to accept this manuscript for publication.

General Comments:

1. Please explain. how do environmental conditions in the season (temperature, light, nutrient availability) influence reproductive phenology and carrageenan biochemical composition of E. isiformis?

We have revised and enriched the manuscript to better explain environmental influences on E. isiformis. In general, factors such as seasonal changes in carrageenan composition and temperature stress can shift carbon allocation from growth to reproduction and modify carrageenan sulfation and polymer length. For example, “seasonal environmental conditions can strongly influence the biochemical composition of carrageenan in E. isiformis by affecting metabolic allocation and cell wall biosynthesis. Reduced light availability during the rainy season may shift carbon allocation from growth toward storage and structural polysaccharides, including carrageenans, while increased nutrient input from freshwater runoff, particularly nitrogen and phosphorus, can enhance photosynthetic efficiency and promote polysaccharide synthesis”. Temperature fluctuations may further modulate enzymatic activity involved in sulfate esterification, influencing the degree of sulfation of carrageenan. Together, these factors likely contribute to observed seasonal variation in carrageenan yield and sulfate content among morphotypes, with peak production occurring during periods of lower irradiance and higher nutrient availability.

Regarding reproductive phenology, despite pronounced morphological and anatomical variability, this variation may reflect phenotypic plasticity influenced by local environmental conditions. Differences in microhabitat characteristics (e.g., hydrodynamics, light exposure, or substrate) could contribute to morphological diversity without detectable molecular divergence. As observed in other tropical seaweeds, “seasonal temperature stress, such as cooler upwelling periods or unusually warm phases often triggers reproductive events, including formation of reproductive structures or increased fragmentation. Extremely high temperatures may suppress gametogenesis, whereas cooler seasonal temperatures are frequently associated with synchronized reproductive phases”.

2. Do the morphotypes in the collection site adapt to different microhabitats and is possibly the reason of morphological differences but not molecular divergence?

As requested, we have clarified the phylogenetic tree (Figure 1) based on concatenated cox1, rbcL, and RuBisCO spacer data. The tree clearly shows that the three identified morphotypes do not form discrete genetic lineages but are instead distributed across shared haplotypes (e.g., Hap01 and Hap02). In response to the reviewer’s suggestion, we have added a discussion of the implications of this finding. We propose that the combination of high morphological variability and genetic uniformity reflects phenotypic plasticity rather than cryptic speciation, indicating that E. isiformis can adjust its morphology in response to localized environmental conditions (e.g., wave exposure or light gradients), a trait commonly observed in other tropical red algae.

3. How does the change of the carrageenan sulphate content of the morphotypes

have any functional value, especially under the conditions of use in the industry?

In general, nutrient stress enhances carrageenan accumulation but alters its chemistry toward more highly sulfated forms. For example, temperature stress, whether cooler or excessively high, produces carrageenan with higher sulfation but reduced gel strength. Conversely, stable and favorable conditions promote growth and yield carrageenan with better gel properties. Seasonal stress conditions tend to shift energy allocation toward reproduction, resulting in more highly sulfated, lower-quality carrageenan. For practical and industrial applications, understanding these changes is critical for cultivation management and harvesting, although in situ modification of seasonal environmental conditions remains challenging.

Specific comments:

1. Abstract suggest to reduce and emphasize the key findings for clarity. The abstract was reduced from 255 to 227 words, emphasizing the key findings.

2. Line 15 and 16: “biomass fitness” → “biomass decline,” “lack of genetic exchange” → “limited genetic exchange.” Suggested word changes: The paragraph was modified.

3. Line 38- 49: Repetition of “genetic diversification.” Please check once. The paragraph was modified.

4. Line 49- 56: Some sentences appear long and complex, which can short sentences to improve clarity. All the manuscript text was reviewed and modified accordingly.

5. Proofread the entire paper and maintain abbreviations (e.g., Eucheumatopsis isiformis abbreviated to E. isiformis). All species abbreviations in the manuscript were reviewed. The abbreviation for Eucheumatopsis isiformis was retained, while species of Eucheuma that are mentioned explicitly were left unabbreviated to avoid confusion.

6. Line 114: The manuscript alternates between referring to the species as Eucheumatopsis isiformis and its synonym Eucheuma isiforme. Consistent usage throughout would avoid confusion. Same as above, all species abbreviations in the document were reviewed. Eucheumatopsis isiformis abbreviation was retained.

7. Line 81 to 211: Detailed technical protocols are included but use subheadings to break down the methods into separate sections (e.g., DNA extraction, PCR, carrageenan extraction) to improve readability. The subheadings in the Methods and Results sections were reorganized to follow a logical and sequential order.

8. Table 3 contains dense data but lacks descriptive legends that clarify significance. Table 3 was deleted, and repetitive numerical details were reduced by presenting reproductive character data as mean ± standard error, condensing our observations in Table 1. Relevant background information from available quantitative records was incorporated into the Discussion section.

9. Sections sometimes jump suddenly from one topic to another (e.g., molecular analysis to carrageenan extraction). The subheadings in the Methods and Results sections were reorganized to follow a logical and sequential order.

Reviewer #2:

This manuscript presents valuable baseline information on the reproductive biology, biochemical composition, and cultivation potential of the red alga Eucheumatopsis isiformis collected from Yucatán, Mexico. The topic is both relevant and timely, as it contributes to the understanding of native eucheumatoid species and their potential applications in sustainable aquaculture, an area of increasing scientific and economic importance.

The study integrates multiple aspects including field sampling, morpho-anatomical characterization, molecular analysis, and carrageenan composition, offering a comprehensive foundation for further research on E. isiformis.

However, while the study is scientifically significant, the manuscript requires structural refinement, stronger statistical justification, and improved consistency in data presentation to meet publication standards. My detailed comments are summarized below for your consideration.

Major Comments

1. Lack of variability reporting in polyamine-induced sporulation experiment

In the results of th

---

## [Decision Letter · Decision Letter 1]

24 Mar 2026

Cultivation potential of the tropical carrageenophyte Eucheumatopsisisiformis (Solieriaceae, Rhodophyta) from Yucatán, Mexico

PONE-D-25-36873R1

Dear Dr. Daniel Robledo

We’re pleased to inform you that your manuscript has been judged scientifically suitable for publication and will be formally accepted for publication once it meets all outstanding technical requirements.

Kind regards,

Inbakandan Dhinakarasamy, Ph.D

Academic Editor

PLOS One

Additional Editor Comments (optional):

Thank you for your careful revision and for satisfactorily addressing the reviewers’ comments. The manuscript is now considered suitable for acceptance; however, we kindly request that a few minor issues be addressed prior to final acceptance. Specifically, please ensure that the references are presented in a consistent format throughout, that all figures (particularly photographs) include clearly visible scale bars, and that a small number of recent and relevant references are incorporated while keeping the overall reference list concise. We would appreciate it if you could make these final adjustments and resubmit the revised manuscript at your earliest convenience to facilitate acceptance without further review.

Reviewers' comments:

Reviewer's Responses to Questions

**Comments to the Author**

Reviewer #1: (No Response)

Reviewer #3: All comments have been addressed

2. Is the manuscript technically sound, and do the data support the conclusions?

Reviewer #1: (No Response)

Reviewer #3: Partly

3. Has the statistical analysis been performed appropriately and rigorously?

Reviewer #1: (No Response)

Reviewer #3: Yes

4. Have the authors made all data underlying the findings in their manuscript fully available?

Reviewer #1: (No Response)

Reviewer #3: Yes

5. Is the manuscript presented in an intelligible fashion and written in standard English?

Reviewer #1: (No Response)

Reviewer #3: Yes

Reviewer #1: The authors answered to satisfy the comments; however, there are a few minor corrections that should be done before acceptance of this manuscript; particularly, the references should be uniform. Figures (photos) should show the scale clearly visible. Latest references should be added and number of references should be limited.

Reviewer #3: 1. What are the challenging processes to extract carrageenan from seaweeds?

2. What percentage of carrageenan yield comes from Eucheumatopsis isiformis?

3. How do you differentiate carrageenan from E. isiformis compared with Kappaphycus alvarezii?

.

Reviewer #1: No

Reviewer #3: No

---

## [Editor Report · Acceptance letter]

PONE-D-25-36873R1

PLOS One

Dear Dr. Robledo,

I'm pleased to inform you that your manuscript has been deemed suitable for publication in PLOS One. Congratulations! Your manuscript is now being handed over to our production team.

Kind regards,

on behalf of

Dr. Inbakandan Dhinakarasamy

Academic Editor

PLOS One